# Improved BDS-2/3 Satellite Ultra-Fast Clock Bias Prediction Based with the SSA-ELM Model

**DOI:** 10.3390/s23052453

**Published:** 2023-02-22

**Authors:** Shaoshuai Ya, Xingwang Zhao, Chao Liu, Jian Chen, Chunyang Liu

**Affiliations:** 1School of Geomatics, Anhui University of Science and Technology, Huainan 232001, China; 2Key Laboratory of Aviation-Aerospace-Ground Cooperative Monitoring and Early Warning of Coal Mining-Induced Disasters of Anhui Higher Education Institutes, Anhui University of Science and Technology, KLAHEI (KLAHEI18015), Huainan 232001, China; 3Coal Industry Engineering Research Center of Mining Area Environmental and Disaster Cooperative Monitoring, Anhui University of Science and Technology, Huainan 232001, China

**Keywords:** ultra-fast satellite clock bias, sparrow search algorithm, extreme learning machine, precision, Beidou satellite navigation system

## Abstract

Ultra-fast satellite clock bias (SCB) products play an important role in real-time precise point positioning. Considering the low accuracy of ultra-fast SCB, which is unable to meet the requirements of precise point position, in this paper, we propose a sparrow search algorithm to optimize the extreme learning machine (SSA-ELM) algorithm in order to improve the performance of SCB prediction in the Beidou satellite navigation system (BDS). By using the sparrow search algorithm’s strong global search and fast convergence ability, we further improve the prediction accuracy of SCB of the extreme learning machine. This study uses ultra-fast SCB data from the international GNSS monitoring assessment system (iGMAS) to perform experiments. First, the second difference method is used to evaluate the accuracy and stability of the used data, demonstrating that the accuracy between observed data (ISUO) and predicted data (ISUP) of the ultra-fast clock (ISU) products is optimal. Moreover, the accuracy and stability of the new rubidium (Rb-II) clock and hydrogen (PHM) clock onboard BDS-3 are superior to those of BDS-2, and the choice of different reference clocks affects the accuracy of SCB. Then, SSA-ELM, quadratic polynomial (QP), and a grey model (GM) are used for SCB prediction, and the results are compared with ISUP data. The results show that when predicting 3 and 6 h based on 12 h of SCB data, the SSA-ELM model improves the prediction model by ~60.42%, 5.46%, and 57.59% and 72.27%, 44.65%, and 62.96% as compared with the ISUP, QP, and GM models, respectively. When predicting 6 h based on 12 h of SCB data, the SSA-ELM model improves the prediction model by ~53.16% and 52.09% and by 40.66% and 46.38% compared to the QP and GM models, respectively. Finally, multiday data are used for 6 h SCB prediction. The results show that the SSA-ELM model improves the prediction model by more than 25% compared to the ISUP, QP, and GM models. In addition, the prediction accuracy of the BDS-3 satellite is better than that of the BDS-2 satellite.

## 1. Introduction

The atomic clocks on satellites provide a reference time for the satellites. As a result, the study of satellite clock bias (SCB) has been a focus of the research community [1,2,3]. Currently, the most precise clock product of IGS has a lag of ~12 days and an accuracy of 75 ps. These characteristics meet the accuracy requirements of precise single-point positioning. However, the prediction accuracy of the ultra-fast clock product is ~3 ns. Please note that a SCB as small as 1 ns produces a distance error of approximately 0.3 m [4]. Therefore, improving the prediction accuracy of clock bias for real-time positioning is of great significance.

Various scholars have been working on SCB prediction models. The well-known and commonly used SCB prediction models include the quadratic polynomial (QP) model [5], the grey model (GM) [6], the autoregressive integrated moving average (ARIMA) model [7], and the Kalman filter (KF) model [8]. These models achieve good performance and are widely applied in SCB prediction. Based on the obvious periodicity of SCB, the prediction accuracy is improved by adding the periodic term when using the QP model to predict SCB [9,10]. Wang et al. proposed a combination model based on the QP model in addition to the periodic term, which used the ARIMA model to fit and predict the residuals of the QP model. The predicted values are added to obtain the final SCB prediction value, which has better accuracy. GM can accurately predict based on a few samples [11]. For the selection of GM parameters, the swarm intelligence optimization algorithm can also be used, in addition to the least square method [12,13,14]. Mei et al. used the order ratio sequence of SCB for GM prediction. The consecutive multiday predictions showed that the SCB of BDS accuracy was improved by more than half as compared with the conventional GM [15]. Liang et al. conducted SCB prediction based on the single difference, which effectively improved the prediction accuracy and stability of SCB [16]. Song et al. introduced a sage window adaptive adjustment residual covariance matrix to improve the KF model’s prediction accuracy [17]. Considering the shortcomings of the single model, the combined model can be divided into two parts, i.e., the prediction model that refits the residual value and the addition of weight to different prediction models [18,19,20,21,22]. Yu et al. used the GM to fit the residual values of the exponential smoothing (ES) model. Finally, they added the prediction values of the GM and ES model to obtain the final SCB prediction value [19]. Li et al. proposed the optimal non-negative variable weight strategy to address the excessive deviation of a single model prediction [20].

Considering the characteristics of nonlinear variations in SCB, machine learning algorithms can be effectively used for its approximation. Wang et al. used a wavelet neural network to predict the single difference in the SCB and compared the results with QP and GM models. The results showed that the prediction effect of the neural network was significantly better [23]. Zhu et al. used least squares support vector machine to predict the SCB of the PHM clock. The experimental results show that the model is significantly better than the linear model [24]. Due to the random generation of parameters in machine learning algorithms, the prediction results are unstable. The swarm intelligence optimization algorithm can effectively address this problem. Wang et al. used a particle swarm optimization wavelet neural network to predict the short-term SCB of GPS. The accuracy obtained using this method was 0.3 ns better within 1 h, which is approximately 75% higher than the traditional model [25]. Lu et al. optimized a BP neural network based on the evolution algorithm and improved the adaptive ability and global search ability of the BP neural network [26]. This model achieves good prediction performance in the short term. Therefore, machine learning algorithms can effectively improve the accuracy of SCB prediction.

The models mentioned above focus on the precision of SCB. However, with the continuous development of society, people’s requirements for real-time positioning accuracy are constantly increasing. As the accuracy of clock offset seriously affects the positioning, research regarding real-time clock offset is of great significance. Lu et al. used a sliding window mode to improve the QP model with an additional period term, then performed SCB prediction for the IGS ultra-fast clock (IGU) product. The accuracy of this model is significantly better compared with the prediction of IGU (IGU-P) [27]. Huang et al. optimized the QP model by changing the weight of SCB and additional period terms, thereby improving the prediction accuracy of IGU-P within 24 h [28]. Li et al. optimized the parameters of the kernel extreme learning machine based on the global optimization ability of the particle swarm optimization algorithm. The authors used the SCB of the iGMAS ultra-fast (ISU) product to analyze the prediction accuracy of satellites with different orbits [29]. The analysis mentioned above focuses on the ultra-fast SCB prediction of GPS and BDS-2 systems, while research on the ultra-fast SCB prediction of the BDS-3 system is relatively less extensive.

Since the BDS-3 started providing navigation and timing services to global users on 31 July 2020, it is necessary to analyze the SCB prediction of the BDS-3 satellite. Currently, GFZ, IGS, and iGMAS provide BDS-3 ultra-fast SCB products. iGMAS was initiated by China in 2012, with 30 tracking stations and 13 analysis centers. The main purpose of iGMAS is to establish a global near-real-time tracking network with ACs and multirepeat coverage of BDS/GPS/GLONASS/Galileo [30]. Since the BDS-3 and iGMAS were built relatively late, little research is available regarding their service performance. Therefore, in this work, we propose a sparrow search algorithm to optimize the extreme learning machine (SSA-ELM) algorithm to predict the SCB. Based on the global optimization ability of the sparrow search algorithm, the initial weights and thresholds of the extreme learning machine are determined. Afterward, the initial parameters are substituted into the extreme learning machine for training and learning. Finally, the trained network model is used for SCB prediction. In this work, the observation data of ISU are used for modeling and prediction, and the accuracy of ISU-P, QP, GM, and SSSA-ELM models is analyzed.

The rest of the paper is organized as follows. In Section 2, we present some basic principles of the SSA-ELM model. Section 3 compares and analyzes ultra-fast clock product data analysis and prediction accuracy of GM, the QP model, ISUP, and the SSA-ELM model. In Section 4, an analysis of multiday clock bias prediction is provided. Finally, we conclude this work in Section 5.

## 2. Basic Principles

### 2.1. Extreme Learning Machine Algorithm

Extreme learning machine (ELM) is an algorithm for machine learning based on feedforward neural networks [31]. ELM is commonly used to train feedforward neural networks with a single hidden layer. An input layer, a hidden layer, and an output layer are included in the network structure. The idea is to randomly generate the weights and thresholds from the input layer to the hidden layer. These weights and thresholds do not need to be updated during training, which drastically reduces training time. The output value of the hidden layer is then calculated using the activation function of the hidden layer. The Moore–Penrose (MP) generalized inverse matrix [32,33,34] is used to obtain the hidden layer output matrix based on the principle of minimizing the difference between the output value of the network and the expected output value. The network structure of ELM is shown in Figure 1.

As presented in Figure 1, $X=[x_{1},x_{2},\cdots ,x_{n}]$ denotes the input data vector, $Y=[y_{1},y_{2},\cdots ,y_{n}]$ denotes the output data vector, $W=[w_{1},w_{2},\cdots ,w_{n}]$ denotes the weight vector from the input layer to the hidden layer, $B=[b_{1},b_{2},\cdots ,b_{n}]$ denotes the bias term vector from the input layer to the hidden layer, $H=[h_{1},h_{2},\cdots ,h_{n}]$ denotes the output value vector of the hidden layer, β denotes the weight array from the hidden layer to the output layer, and $Y=[y_{1},y_{2},\cdots ,y_{n}]$ denotes the expected output vector. The main process of the ELM algorithm is presented below.

First, based on the randomly generated weights and thresholds, the output value of the hidden layer is calculated using the activation function of the hidden layer. This is mathematically expressed as follows:(1)h(x)=g(w⋅x+b)
where g(x) denotes the activation function, w denotes the initial weight, b denotes the initial bias term, and h(x) denotes the output value of the hidden layer.

The output matrix of the hidden layer and the expected output value of the output layer are used to calculate the output weight matrix of the hidden layer according to the principle of minimum error. The output matrix is expressed as follows:(2)Y=Hβ
(3)$$H(w_{1},\cdots ,w_{L},b_{1},\cdots ,b_{L})=\left[\begin{array}{l} g(w_{1}\cdot x_{1}+b_{1})\;\;\;\cdots \;\;\;g(w_{L}\cdot x_{1}+b_{L})\\ \;\;\;\;\;\;\;\;\vdots \;\;\;\;\;\;\;\;\;\;\;\;\;\;\;\;\;\;\;\;\;\;\;\;\;\;\;\;\vdots \\ g(w_{1}\cdot x_{N}+b_{1})\;\;\;\cdots \;\;g(w_{L}\cdot x_{N}+b_{L}) \end{array}\right]$$
where H denotes the output matrix of the hidden layer, Y denotes the expected output value, and β denotes the output weight matrix of the hidden layer. The minimum norm is used to solve the optimal value and is expressed as follows:(4)$$\beta =H^{+}Y$$
where $H^{+}$ denotes the transformation matrix of the Moore–Penrose generalized inverse of the matrix.

### 2.2. Sparrow Search Algorithm

The sparrow search algorithm is an optimization algorithm based on swarm intelligence. It is mainly based on the foraging process of sparrows to find the best solution. To accomplish this process, the sparrows are divided into producers and followers. The producers are responsible for locating food, and the followers follow them to obtain food. When sparrows become aware of a threat, a portion of the population becomes vigilant and engages in antipredator behavior [35,36,37]. The main update process is presented below.

(1) Since the initial population is randomly generated and the individual position distribution is not uniform, the accuracy of the resulting solution is not high. Therefore, an improved Henon chaotic map is used to initialize the sparrow population. This is mathematically expressed as follows:(5)$$\begin{array}{l} x_{n+1}=-ax_{n}^{2}+\left|y_{n}\right|+1\\ y_{n+1}=bx_{n} \end{array}$$
where $y_{n+1}$ represents the initial sparrow population obtained using the Henon chaotic map, a=0.6, and b=0.7.

(2) To enhance the global optimization and local mining ability of the algorithm, the inertia weight is added to the finder, and the final producer’s position update formula is expressed as follows:(6)$$X_{i,j}^{t+1}=\left\{\begin{array}{l} X_{i,j}^{t}\cdot w\cdot \exp\left(\frac{-i}{\alpha \cdot t_{\max}}\right),\;\;if\;\;R_{2}<S_{ST}\\ X_{i,j}^{t}+Q\cdot L,\;\;\;\;\;\;\;\;\;\;\;\;\;if\;\;R_{2}\geq S_{ST}\; \end{array}\right.$$
(7)$$w=w_{\max}-\frac{w_{\max}-w_{\min}}{t_{\max}^{2}}\cdot t_{}^{2}$$
where t represents the number of iterations, $X_{i,j}^{t}$ denotes the latitude (j) of the i-th sparrow during the t-th iteration, $t_{\max}$ denotes the maximum number of iterations, α denotes a random number between (0, 1), $R_{2}$ denotes a random number between (0, 1), $S_{ST}$ denotes a random number between [0.5, 1], L denotes a matrix of dimension 1×d (all the elements in this matrix are 1), d denotes the dimension of a sparrow, Q denotes a random number between (0, 1), $w_{\max}$ denotes the maximum weight, and $w_{\min}$ denotes the minimum weight. The number of finders generally represents 10 to 20 percent of the total population.

(3) The follower’s location is updated according to the following expression:(8)$$X_{i,j}^{t+1}=\left\{\begin{array}{l} Q\cdot \exp\left(\frac{X_{worst}^{t}-X_{i,j}^{t}}{i^{2}}\right),\;\;\;\;\;\;\;\;\;\;\;\;if\;\;i>\frac{n}{2}\\ X_{p}^{t+1}+\left|X_{i,j}^{t}-X_{p}^{t+1}\right|\cdot A^{+}\cdot L,\;\;if\;\;i\leq \frac{n}{2}\; \end{array}\right.$$
where $X_{p}$ denotes the current optimal position occupied by the discoverer, $X_{worst}$ represents the current global worst position, A represents a matrix of dimension 1×d in which each element is randomly assigned values of 1 or −1, $A^{+}=A^{Τ}{\left(AA^{Τ}\right)}^{-1}$, and n denotes the number of sparrows.

(4) When the sparrow population is aware of vigilantes, they showcase antipredation behavior. The vigilant location update formula is expressed as follows:(9)$$X_{i,j}^{t+1}=\left\{\begin{array}{l} X_{best}^{t}+\beta \cdot \left|X_{i,j}^{t}-X_{best}^{t}\right|,\;\;\;if\;\;f_{i}>f_{g}\;\;\\ X_{i,j}^{t}+K\cdot \left(\frac{\left|X_{i,j}^{t}-X_{worst}^{t}\right|}{(f_{i}-f_{w})+Ɛ}\right),\;\;if\;\;f_{i}=f_{g}\; \end{array}\right.$$
where $X_{best}^{t}$ and $X_{worst}^{t}$ represent the global optimal and worst positions during the t-th iteration, respectively; $f_{i}$ denotes the fitness value of the current individual sparrow; β denotes the parameter of the step size; K denotes a random number in the range [−1, 1]; $f_{g}$ and $f_{w}\;$ represent the global best and worst fitness values, respectively; and ε represents a small constant to avoid zero in the denominator.

### 2.3. SSA-ELM Model

Since ELM does not need to update the weights and thresholds of the input and hidden layers during the training process, its training time is significantly reduced compared to the BP neural network. In addition, the initial weights and thresholds are generated randomly, so they significantly impact the prediction accuracy of ELM. To address this issue, SSA is used for optimization. In this study, we optimize the ELM to predict the SCB using the SSA. The main process is divided into two parts: (1) The SSA optimizes the ELM’s initial parameters. (2) The SSA-estimated optimal initial values are substituted in the ELM for training. Figure 2 depicts the operation of the SSA-ELM model:

Step 1. The number of hidden layer nodes in the ELM is determined based on the input and output values of the network.

Step 2. The Henon chaotic map is used to initialize the SSA algorithm. 

Step 3. The fitness function is used to calculate the fitness value of each sparrow, and the sparrow population is divided into producers and followers, accounting for 20% and 80% of the population, respectively.

Step 4. The positions of the producers are updated using Expression (6), and random values of $R_{2}$ are used as the discriminant condition of Expressions (6), i.e., $S_{ST}=0.7$.

Step 5. The positions of the followers are updated using Expression (8), and the median of the sparrow population is used as the discriminant condition of Expression (8).

Step 6. The positions of the vigilantes are updated using Expression (9), with the proportion of vigilance accounting for 10%, which is randomly generated in the sparrow population. The optimal fitness value is used as the discriminant condition of Expression (9).

Step 7. The optimal parameters obtained by the SSA algorithm are substituted into the ELM for continuous training and prediction.

## 3. Ultra-Fast Clock Bias Data Analysis

In this work, we selected the ultra-fast clock products for 14 days (DOY 196–210, 2022) from iGMAS to assess the quality of the SCB. The observational data anomaly of the ultra-fast clock product occurring on day 207 is not investigated. The ultra-fast clock product contains 48 h of data, with the first 24 h being observation data (ISUO) and the last 24 h being prediction data (ISUP), with a 15 min sampling interval. The precision of ISUO data is 1 ns. Please be aware that the ISUP data are extrapolated from the ISUO data, and their precision is 10 ns. The ultra-fast clock product is updated every 6 h, i.e., at the outset and at the 6th, 12th, and 18th hours. In addition, the ISUO data have a three-hour lag time, whereas the ISUP data are real-time. In the experiments, the precision clock product (ISC) is considered the ground truth, and the root mean square (RMS) and standard deviation (STD) of the residual values are used to evaluate the accuracy and stability of the prediction [38,39]. This is expressed mathematically as follows:(10)$$RMS=\sqrt{\frac{1}{n}\sum_{i=1}^{n}{\left(x_{i}-y_{i}\right)}^{2}}$$
where n denotes the number of samples, $x_{i}$ denotes the clock bias data, and $y_{i}$ denotes the true value of the clock bias.
(11)$$STD=\sqrt{\frac{1}{n-1}\sum_{i=1}^{n}{\left(\Delta_{i}-\mu \right)}^{2}}$$
where $\Delta_{i}$ denotes the residual between the clock bias data and the true value, and μ denotes the mean value of the residual.

As different reference clocks are selected to calculate the precision clock product and ultra-fast clock products, it is necessary to select a satellite with good data quality as the reference clock. This ensures the difference between other satellites and the reference clock and eliminates the errors of the system [28,40,41,42]. In this work, all the satellites of the BDS are tested. The details of the satellites are shown in Table 1. Since the atomic clock of BDS-2 is based on the Rb clock and the atomic clock of BDS-3 is based on the new rubidium (Rb-II) clock and hydrogen (PHM) clock, the experiment uses single-day and multiday data to analyze the accuracy and stability of different atomic clocks.

### 3.1. Single-Day Accuracy Analysis

In this work, we use the ultra-fast clock products of the day (DOY 196, 2022) for analysis. The ultra-fast clock product updated at the outset and the 6th hour is selected for the experiments, and the first 6 h of daily data are used as the experimental data. In addition, C11 (Rb clock), C20 (Rb-II clock), and C27 (PHM clock) are selected as reference clocks, and the three residual sequences obtained by the second difference are ISUP-ISC, ISUO-ISC, and ISUP-ISUO. We select different reference clocks, i.e., C08 (Rb clock), C22 (Rb clock), and C29 (PHM clock), for precision analysis, as shown in Figure 3, Figure 4 and Figure 5.

As presented in Figure 3, when the Rb clock is used as the reference clock, the initial residual values of the three residual sequences of C08 are different, and the absolute value of the initial residual value of ISUP-ISC is the lowest. The absolute values of the initial residual values of ISUP-ISC and ISUO-ISC in the three residual sequences of C22 and C29 are lower than that of ISUP-ISUO. In addition, please note that the fluctuations in the three residual sequences of C22 and C29 are similar, in contrast to those of C08.

As shown in Figure 4, when the Rb-II clock is used as the reference clock, the initial values of the three residual sequences of C22 and C29 do not differ significantly, and the initial residual values of ISUP-ISC and ISUO-ISC are approximately 1 ns. In contrast, the initial residual value of ISUP-ISUO is approximately 0. However, the initial values of the three residual sequences of C08 are relatively large, and the absolute value of the initial residual value of ISUP-ISUO is the lowest. Figure 5 demonstrates that the residual sequence using the PHM clock as the reference clock is consistent with Figure 4.

The analysis of different residual values shows that the initial residual values of ISUP-ISUO for the Rb-II clock and PHM clock are lower than those for the Rb clock. Moreover, the value does not change with different reference clocks. The initial residual values of ISUP-ISC and ISUO-ISC vary greatly when the reference clock is different. For example, when C22 uses the Rb clock as the reference clock, the initial residual values of ISUP-ISC and ISUO-ISC are about −5 ns, and when the Rb-II clock and PHM clock are used as reference values, the initial residual values are 1 ns and −1 ns, respectively. In addition, when the Rb-II clock and PHM clock are used as the reference clocks, the three kinds of residual sequences of the SCB change minutely. On the contrary, when the Rb and Rb-II clock and the Rb and PHM clock are used as the reference clocks, the residual sequences of the SCB change significantly. Table 2 shows the average accuracies of all the satellites with different reference clocks.

As shown in Table 2, the analysis of various reference clocks reveals that when C11 is used as the reference clock, the rubidium clock’s accuracy is 2.39 ns, which is greater than 4.05 ns and 3.23 ns for the Rb-II and PHM clocks, respectively. When C22 is used as the reference clock, the Rb-II clock’s accuracy is 0.73 ns, which is greater than 4.58 ns and 1.42 ns for the Rb-II and PHM clocks, respectively. The precision of the PHM clock is 0.68 ns when C27 is used as the reference clock, which is greater than 3.22 ns and 1.33 ns for the Rb-II and PHM clocks, respectively. When the reference clock is different, the precision of various types of atomic clocks varies greatly.

ISUP-ISUO has the highest accuracy, while ISUP-ISC has the lowest accuracy, which is unaffected by the reference clock. In addition, the accuracies of ISUP-ISC and ISUO-ISC vary considerably when different reference clocks are used. When C20 and C27 are used as reference clocks, the precision of various types of atomic clocks varies slightly. However, hen C11 is used as a reference clock, the accuracies vary significantly.

### 3.2. Multiday Accuracy and Stability Analysis

To increase the effectiveness of the analysis, we use 13 days (DOY 197–210,2022) of ultra-fast clock products for accuracy and stability analysis. We select satellites with better data quality as the reference clocks. Table 3 shows the average accuracy of different types of atomic clocks 6 h before 13 days.

As presented in Table 3, when the rubidium clock is selected as the reference clock, the hydrogen clock’s precision is approximately 3.82 ns. When the new rubidium clock is selected as the reference clock, its accuracy is approximately 0.91 ns. When the hydrogen clock is selected as the reference clock, its precision is approximately 1.09 ns. This demonstrates that the reference clock significantly affects the precision of various atomic clocks. For Rb-II and PHM clocks, the accuracy is superior to that of other types of atomic clocks when the reference clock is of the same type. No matter which reference clock is chosen for the rubidium clock, its accuracy is at its lowest. Comparing various residual sequences, ISUP-ISC has the lowest overall accuracy, while ISUP-ISUO has the highest. This result is consistent with the analysis of a single day.

Based on an analysis of the change in the degree of accuracy caused by different reference clocks, the average change in the accuracy of the three atomic clocks is relatively minimal. However, when Rb and Rb-II or Rb and PHM are used as the reference clocks, the changes in the accuracies of the atomic clocks are greater. Therefore, changes in the system of reference clocks cause a substantial change in the accuracy of satellite clock bias. Moreover, Rb-II and PHM clocks are part of the BDS-3 system, and the SCB precision of BDS-3 is superior to that of BDS-2.

The residual value of the second difference of SCB reflects stability. This study employs STD to evaluate stability. Since stability is unrelated to system error, the Rb clock’s 13-day residual value is chosen as the reference clock for analysis. Figure 6, Figure 7 and Figure 8 depict the distribution of residual values for ISUP-ISC, ISUO-ISC, and ISUO-ISUO. The average STD values of various atomic clocks are shown in Table 4.

As presented in Figure 6, the residuals of the Rb clock are concentrated between −10 ns and 10 ns. Please note that these data are more dispersed as the residuals of the Rb-II clock are concentrated between −4 ns and 0 ns. The residual values of the PHM clocks are distributed between −2 ns and 1 ns, which basically conforms to the standard normal distribution. Therefore, the residual concentration of PHM clocks is better than that of the Rb and Rb-II clocks. Figure 7 shows that the residuals of ISUO-ISC are similar to those of ISUP-ISC. 

As presented in Figure 8, the residual values of ISUP-ISUO show standard normal distribution. Moreover, the concentration degree of the three atomic clocks is better than that of ISUP-ISC and ISUO-ISC. The analysis of different atomic clocks shows that the distribution interval of the Rb clock is the largest, and the concentration degree is lower than that of Rb-II and PHM clocks. The concentration degree of the PHM clock is high. The STD mean values of different atomic clocks in the residual sequence are presented in Table 4.

As presented in Table 4, the residual value of ISUO-ISC is optimal for STD values of different atomic clocks. In this case, the STD values of Rb, Rb-II, and PHM clocks are 2.78 ns, 0.73 ns, and 0.63 ns, respectively. The residual value of ISUP-ISC is the worst for STD values of different atomic clocks. In this case, the STD values of the Rb, Rb-II, and PHM clocks are 3.71 ns, 1.41 ns, and 0.99 ns, respectively. Therefore, the ISUO-ISC is the most stable. Moreover, the mean value of the PHM clock is 0.84 ns, which is superior to the 3.15 ns and 1.06 ns mean values of the Rb and Rb-II clocks, respectively. Therefore, the PHM clock is the most stable, followed by the Rb-II clock, and the Rb clock is the least stable. Therefore, the stability of BDS-3 is better than that of BDS-2.

## 4. Accuracy Analysis of Ultra-Fast Clock Bias Prediction

This work uses the ultra-fast clock bias files published by iGMAS in the DOY of 196–204 in 2022 for a total of 7 days to perform the experiment; the 202nd and 203rd days are not analyzed due to the abnormality of the data. The SCB is predicted using the QP, GM, and SSA-ELM models, and the models’ SCB accuracy are compared to that of ISUP. In this work, the observation part of the ultra-fast SCB is regarded as the ground truth, and the accuracy of the predicted SCB of the model is determined by computing the difference between the predicted SCB and the ground truth. This work involves the analysis of single-day and multiday forecasts. Due to the updating time and lag time of ultra-fast SCB, this study focuses on the accuracy of SCB within 6 h. Since the ultra-fast clock bias generates four files per day, the data overlap in different time periods. In this work, the observation clock bias in for the initial update file is used as the fitting data. For comparative analysis, the observation data from the 6th hour of the same day and the initial hour of the following day are used as the true values.

### 4.1. Prediction Analyses of Clock Bias Data with Different Fitting Times

In the experiment, the observation data for the 2022 DOY 196 ultra-fast clock product are selected as the fitting data, and the observation data of the next day are used as the ground truth for accuracy evaluation. QP, GM, and SSA-ELM models are used to fit 24 h prediction and 6 h clock bias prediction, and the results are compared with ISUP. Figure 9, Figure 10, Figure 11 and Figure 12 show the residual values of ISUP, QP, GM, and SSA-ELM, respectively. Table 5 shows the average accuracy of 6 h prediction based on different atomic clocks. 

As presented in Figure 9, Figure 10, Figure 11 and Figure 12, the residual value of ISUP increases with time, and the residual value of BDS-2 increases faster than BDS-3. Moreover, the trend of the residual value of BDS-2 is complex in terms of rising, falling, and being stationary, and the absolute value of the residual value of some satellites exceeds 10 ns. However, the residual values of BDS-3 show a decreasing trend, ranging from −5 ns to −10 ns. The trends of the residual value of the QP and GM models are similar to ISUP. However, the interval of residual values for the BDS-2 and BDS-3 of the QP model is smaller than that of the ISUP. In addition, the residual value of the GM model for some satellites reaches 100 ns. In the prediction clock bias of the SSA-ELM model, the residual values of some satellites increase rapidly with an increase in the prediction time. It is noteworthy that some satellites are relatively stable, and this situation occurs for BDS-2/BDS-3.

At the same time, it is evident that the initial residual values of ISUP and prediction models are large. The initial BDS-2 residual values range from −2 ns to −8 ns. Please note that the concentration of these values is greatest for the SSA-ELM model. The initial BDS-3 residual values range from −4 ns to −6 ns. Based on this phenomenon in the prediction models, it can be judged whether there has been a large systematic error in the ultra-fast clock product in the past two days and the two upcoming days. In addition, some residuals of the SSA-ELM model increase too rapidly due to an excessive amount of system error.

Aiming at the problem of a large system error caused by the ultra-fast clock product updated in the initial hour the next day as the true value, this work selects the observation data of the ultra-fast clock product updated at the 6th hour of the day as the true value for experimental analysis. For the accuracy analysis of predictions performed at different times, QP, GM, and SSA-ELM models are selected for accuracy analysis of 3 h and 6 h predictions, where the first 24 h of data of the ultra-rapid clock product are used for modeling and are updated in the initial hour of the day.

As presented in Figure 13, Figure 14, Figure 15 and Figure 16, when the clock product updated at the 6th hour of the day is used as the true value, the system error can be significantly decreased. The analysis of various models reveals that the residuals of the QP model and ISUP are identical and increase over time. In addition, the ISUP model has a faster rate of growth than the QP model. The residual of the SSA-ELM model is the most concentrated and has the best stability. The residual of BDS-2 ranges between −4 ns and 4 ns, while that of BDS-2 ranges between −2 ns and 2 ns. The GM model has the largest residual range, and some satellite predictions are incorrect. The analysis of various BDS systems reveals that BDS-2 has a much faster residual growth rate and a wider residual distribution range than BDS-3, which may be related to the atomic clock carried by BDS.

As shown in Table 5, when 24 h of data are used to predict the following 3 h of clock bias, the ISUP has the worst prediction accuracy, i.e., 1.75 ns for RB clocks, 1.65 ns for RB-II clocks, and 1.62 ns for PHM clocks. In the same order as ISUP, the QP, GM, and SSA-ELM models rank the prediction accuracy of various atomic clocks. Based on an analysis of the average prediction accuracy of various models, the SSA-ELM model has the best prediction accuracy of 0.51 ns, while the ISUP model has the worst prediction accuracy of 1.67 ns. Moreover, the prediction accuracy of the SSA-ELM model for ISUP, QP, and GM models improved by 60.42%, 5.46%, and 57.59%, respectively. The accuracy improvement in the SSA-ELM model is the highest for ISUP.

Table 6 demonstrates that when 24 h data are used to predict the clock bias for the next six hours, the prediction accuracies of different models is comparable to those listed in Table 5. Please take note that the SSA-ELM model is superior to the ISUP model. ISUP and the aforementioned three prediction models provide the most accurate predictions for the PHM clock. However, they reach different conclusions regarding the least atomic clocks. SSA-ELM, GM, and QP models have the worst prediction accuracy for the Rb clock, and ISUP has the worst prediction accuracy for the Rb-II clock. In addition, the prediction accuracy of the SSA-ELM model for ISUP, QP, and GM models improved by 72.27%, 44.65%, and 62.96%, respectively. On the basis of the comparison presented in Table 5 and Table 6, it is evident that as the prediction time increases, the accuracies of various prediction models decrease to varying degrees. The SSA-ELM model’s prediction accuracy changes the least, while the ISUP model’s prediction accuracy changes the most. Moreover, the prediction accuracy of the SSA-ELM model for 3 h and 6 h SCB prediction is within 1 ns.

At the same time, the prediction accuracy of the SSA-ELM model is the best among the four models. It is notable that the accuracy of this prediction does not change over time. Moreover, the prediction accuracy of the SSA-ELM model changes minutely, and the prediction results are stable. The prediction accuracy of ISUP is the worst, and the prediction of the GM model is unstable.

### 4.2. Clock Bias Prediction with Different Fitting Time Lengths

In order to analyze the effect of prediction time on the prediction accuracy, QP, GM, and SSA-ELM models are used for comparative analysis. Since ISUP is extrapolated from the 24 h ISUO data, the strategies of fitting 6 h and 12 h of SCB data to predict 6 h are used for the experiment. Figure 17 and Figure 18 show the prediction accuracy at different prediction times. Table 7 and Table 8 show the mean values of the prediction accuracies of different atomic clocks. Please note that the anomaly of C10 data is not discussed.

As shown in Figure 17 and Figure 18, the SSA-ELM model’s prediction accuracy is superior to that of the QP and GM models. Moreover, this model exhibits few variations across various satellites. When fitted with 6 h of SCB data, the QP model’s prediction accuracy is within 8 ns for BDS-2 and 4 ns for BDS-3. The GM model’s precision varies greatly between satellites. C04 and C33 have the worst prediction accuracy for BDS-2 and BDS-3 satellites using the GM model, with prediction errors greater than 8 ns. As the fitting time increases, the accuracy of the GM model for C04, C19, and C33 decreases significantly, while the performance of the QP model for RB clocks improves significantly. Figure 6 and Figure 7 illustrate the statistical average accuracy of the three prediction models. Please be aware that the anomaly satellite is not taken into account during calculations.

As presented in Table 7 and Table 8, when fitting the 6 h of SCB data, the average prediction accuracy of the SSA-ELM model is 0.76 ns, that of the QP model is 1.17 ns, and that of the GM model is 1.62 ns. Analysis of the prediction accuracies of different atomic clocks shows that the QP and SSA-ELM models have the worst prediction accuracies for RB clocks, and the GM model has the worst prediction accuracy for Rb-II clocks. Compared to the QP and GM models, the prediction accuracies of the SSA-ELM model improved by 53.16% and 52.09%, respectively. When fitting 12 h of SCB data, the prediction accuracy of the QP model is 1.55 ns, which is improved to a certain extent compared to fitting 6 h of SCB data. However, the prediction accuracies of GM and SSA-ELM models deteriorate constantly. In addition, the prediction accuracy of the SSA-ELM model improves by about 40.66% and 46.38% compared to the QP and GM models, respectively.

The statistical results presented in Table 6, Table 7 and Table 8 demonstrate that when the fitting time varies, all the average prediction accuracies of the SSA-ELM model are within 1 ns, while those of the QP and GM models are within 2 ns. Please note that the prediction accuracy of the SSA-ELM model is 40% higher than that of the QP and GM models. In addition, when SSA-ELM and GM models are used to predict short-term clock bias, the greater the accuracy of the prediction, the worse the fit of the data. The analysis of various atomic clocks reveals that the accuracy of the three prediction models is lowest for the Rb clock and highest for the PHM clock. These precisions are unaffected by the change in fitting time.

### 4.3. Analysis of Multiday Clock Bias Prediction

To verify the analysis results, 6 days of ultra-fast clock bias data are selected. First, the initial residual values of ISUP and ISUO in 6 days are counted. The ISUO data are selected from the file updated at the 6th hour of the day and the initial hour of the next day. The statistics of the average initial residual values of different atomic clocks are presented in Figure 19 and Figure 20 and Table 9. Then, QP, GM, and SSA-ELM models are used to predict the 6 h clock bias, and the data are compared with the ISUP clock bias. Please note that C04 is not analyzed due to poor data quality. The average accuracy of each satellite in the BDS system for 6 days is calculated, and the prediction accuracy of different atomic clocks is analyzed. Figure 21 and Figure 22 show the clock bias prediction accuracy of BDS-2 and BDS-3 satellites, respectively. Figure 23, Figure 24, Figure 25 and Figure 26 show the distribution of clock bias residual values of ISUP, QP, GM, and SSA-ELM models, respectively. Table 10 shows the statistical results of the 6-day average accuracy of different atomic clocks.

As shown in Figure 19 and Figure 20, the absolute value of the initial residual updated at the 6th hour of the day is obviously less than its absolute value at the initial hour of the following day. Please note that this value does not change with different satellites. Therefore, the system error of the ultra-fast SCB observation file updated at the 6th hour of the day is significantly less than that of the file updated at the initial hour of the following day. Additionally, the absolute value of the initial residual varies significantly with DOY. When the DOY is 196, the absolute value is approximately 1 ns, while when the DOY is 198, the absolute value is approximately 10 ns. According to Table 9, system errors exist in various clock bias files. In addition, RB clocks have a larger system error than Rb-II and PHM clocks.

As presented in Figure 21, for BDS-2, the prediction accuracy of the SSA-ELM model is optimal, which is within 6 ns. The maximum prediction accuracy of C10 is also within 9 ns. The prediction accuracy of ISUP and QP models is the same. However, the prediction accuracy of the GM model for different satellites fluctuates greatly. In addition, the prediction accuracy of ISUP, QP, and GM models for C10 and C13 satellites is the worst, exceeding 9 ns. The performance of the SSA-ELM model is significantly higher than that of the other models.

As presented in Figure 22, for BDS-3, the prediction accuracy of the SSA-ELM model remains optimal. The prediction accuracy of different models is basically within 6 ns, except that the prediction accuracy of the GM model for C19, C21, C32, and C33 exceeds 15 ns, which is significantly higher than that for the other satellites. On the other hand, the prediction accuracy of ISUP, QP, and GM models for other satellites is roughly the same.

As presented in Table 10, the worst prediction accuracy of Rb clocks obtained using QP, GM, SSA-ELM, and ISUP is about 6.88 ns, 7.5 ns, 4.54 ns, and 7.08 ns, respectively. In addition, the optimal prediction accuracy of the SSA-ELM model is 4.43 ns, the worst prediction accuracy of the GM model is 6.48 ns, whereas that of the ISUP and QP models is 6.11 ns and 6.23 ns, respectively. As compared with QP, GM, and ISUP models, the SSA-ELM model improves the Rb clocks’ prediction accuracy by about 35.9%, 33.98%, and 39.43%, that of the RB-II clocks by about 20.79%, 23.86%, and 27.72%, and that of PHM clocks by about 23.86%, 28.17%, and 25.72%, respectively. The SSA-ELM model improves the prediction model by about 26.85%, 28.67%, and 30.96% compared to the ISUP, QP, and GM models, respectively. According to the above analysis, it can be concluded that the accuracies of the four clock bias predictions for BDS-3 are better than those for BDS-2. Moreover, the prediction accuracy of the SSA-ELM model is significantly improved as compared with ISUP.

As shown in Figure 23, Figure 24, Figure 25 and Figure 26, the ISUP, QP, and GM models have essentially the same distribution of residual values for various atomic clocks, and the residual values are concentrated. However, the SSA-ELM model’s residual values are scattered. For the Rb clock, the residual values of the ISUP, QP, and GM models range from −15 to 10 ns, mainly centered around 5 and −10 ns. The SSA-ELM model’s residual values range from −10 to 5 ns and are predominantly distributed between 0 and 5 ns. For the Rb-II and PHM clocks, the residual values of the ISUP, QP, and GM models primarily occur between −5 and 10 ns and between 0 and 5 ns. Moreover, the residual value distribution of the SSA-ELM model is similar to that of other atomic clocks. Based on Table 10, it is evident that the cause of the aforementioned phenomenon could be the system error between the observed data and the prediction data of the ultra-fast clock product.

### 4.4. Discussion

In these case studies, we first evaluate the accuracy and stability of iGMAS clock products and conclude that ISUP-ISUO has the highest data accuracy. Consequently, ISUO data are utilized as the actual values for SCB prediction. In addition, the SSA-ELM model’s SCB prediction accuracy is significantly superior to that of the ISUP, QP, and GM models. Moreover, when the fitting time and prediction time differ, the SSA-ELM model’s prediction accuracy is at its highest. Huang et al. and He et al. used the method of second difference to eliminate the systematic error between different clock error products [28,42], making the SCB prediction model more accurate. However, this problem cannot be solved effectively due to its limited real-time performance. The prediction strategy proposed in this paper does not process SCB data, ensuring the data’s authenticity and real-time performance. However, there are some system errors in the SCB prediction, and this research is based on data published by iGMAS, which are subject to network and platform limitations. In the future, we will investigate real-time streaming data and discuss how they affect the precision of precise point positioning.

## 5. Conclusions

This study analyzes the accuracy and stability of prediction for ultra-fast SCB products based on the second difference method. In addition, the accuracy of the ultra-fast SCB at various fitting and prediction times is analyzed using QP, GM, and SSA-ELM models, and the accuracy of the proposed method is compared with that of ISUP.

First, a multiday analysis of ultra-fast clock products and precision clock products reveals that ISUP-ISUO has the highest accuracy and ISUP-ISC has the lowest data consistency. This is because ISUP and ISUO both belong to ultra-fast clock offset files. However, the STD value of ISUO-ISC is the lowest, and the STD value of ISUP-ISC is the highest. The accuracy of the results is also affected by the choice of reference clock. When the BDS-2 satellite is the reference clock, the accuracy is greater than when the BDS-3 satellite is the reference clock, and the same rule applies to BDS-3. PHM clocks have the best STD values, followed by the Rb-II clocks and the Rb clocks.

Second, according to the analysis of the prediction accuracy of the QP, GM, SSA-ELM, and ISUP models, the prediction accuracy of the SSA-ELM model is optimal at different fitting and prediction times. In addition, the prediction accuracy of the SSA-ELM model decreases with increased prediction time, but its accuracy is within 1 ns.

Third, based on the multiday accuracy analysis, the SSA-ELM model improves the prediction model by 26.85%, 28.67%, and 30.96% for the ISUP, QP, and GM models, respectively. In addition, the PHM clock has the highest prediction accuracy among all prediction models. The Rb-II clock and the Rb clock are the second worst, so the prediction accuracy of BDS-3 is superior to that of BSD-2.

The degree of residual concentration of the ISUP, QP, and GM models is better than that of the SSA-ELM model based on the statistical results of different residual values. However, these models have more large residual values than the SSA-ELM model. According to the statistics of various initial residuals, the system error of iGMAS ultra-fast clock bias products is unstable, so the residual values are less distributed around 0.

## Figures and Tables

**Figure 1 sensors-23-02453-f001:**
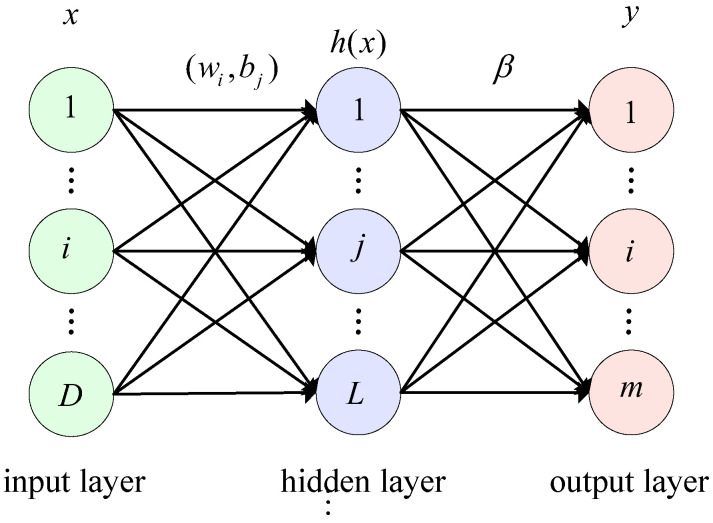
The architecture of ELM.

**Figure 2 sensors-23-02453-f002:**
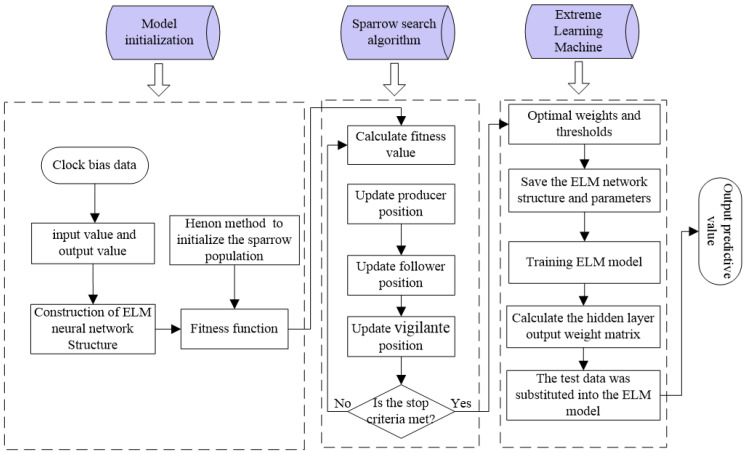
The workflow of the proposed SSA-ELM algorithm.

**Figure 3 sensors-23-02453-f003:**
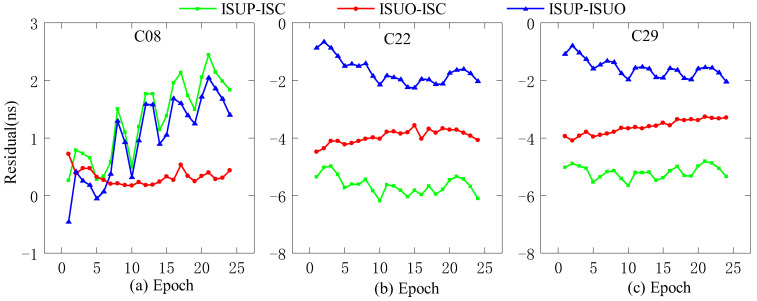
The residual plot of the RB clock as a reference clock. (**a**) The residual value of C08 predicted after 6 h. (**b**) The residual value of C22 predicted after 6 h. (**c**) The residual value of C29 predicted after 6 h.

**Figure 4 sensors-23-02453-f004:**
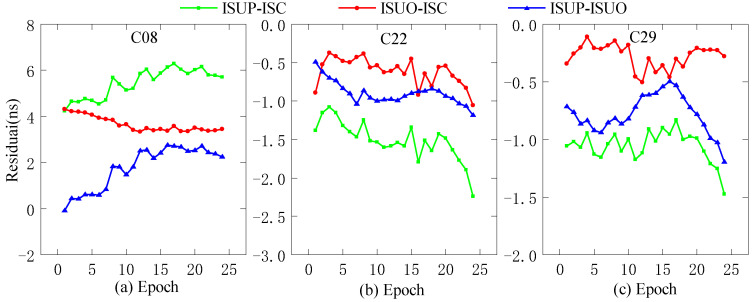
The residual plot of the Rb-II clock as the reference clock. (**a**) The residual value of C08 predicted after 6 h. (**b**) The residual value of C22 predicted after 6 h. (**c**) The residual value of C29 predicted after 6 h.

**Figure 5 sensors-23-02453-f005:**
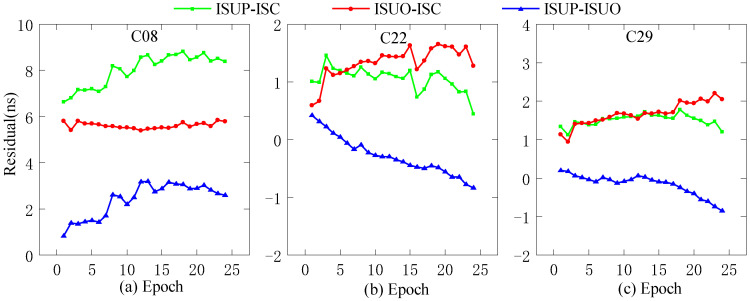
The residual plot of the PHM clock as the reference clock. (**a**) The residual value of C08 predicted after 6 h. (**b**) The residual value of C22 predicted after 6 h. (**c**) The residual value of C29 predicted after 6 h.

**Figure 6 sensors-23-02453-f006:**
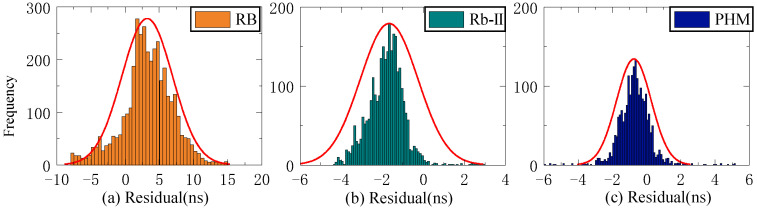
The ISUP-ISC residuals of three atomic clocks. (**a**) The residual distribution of the RB clock. (**b**) The residual distribution of the Rb-II clock. (**c**) The residual distribution of the PHM clock.

**Figure 7 sensors-23-02453-f007:**
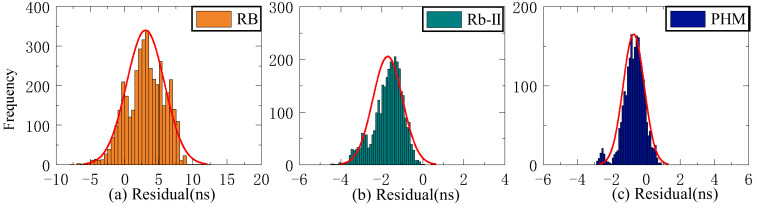
The ISUO-ISC residuals of three atomic clocks. (**a**) The residual distribution of the RB clock. (**b**) The residual distribution of the Rb-II clock. (**c**) The residual distribution of the PHM clock.

**Figure 8 sensors-23-02453-f008:**
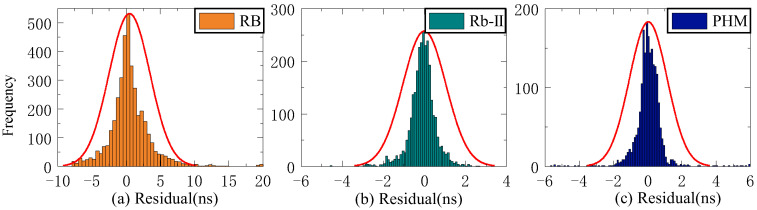
The ISUP-ISUO residuals of three atomic clocks. (**a**) The residual distribution of the RB clock. (**b**) The residual distribution of the Rb-II clock. (**c**) The residual distribution of the PHM clock.

**Figure 9 sensors-23-02453-f009:**
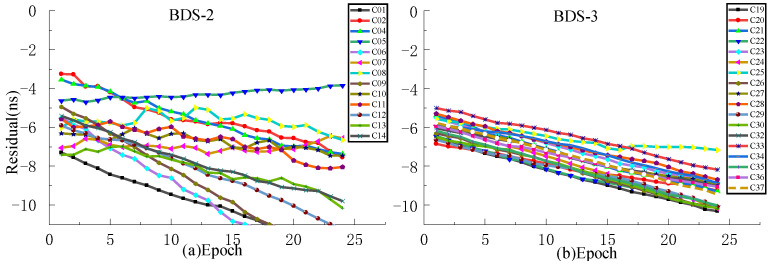
The residual plot of the ISUP model using the initial hour update ultra-fast clock product of the next day. (**a**) The residual value of BDS-2 predicted after 6 h. (**b**) The residual value of BDS-3 predicted after 6 h.

**Figure 10 sensors-23-02453-f010:**
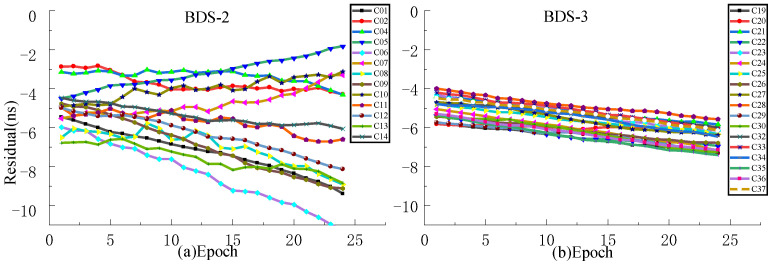
The residual plot of QP model using the initial hour update ultra-fast clock product of the next day. (**a**) The residual value of BDS-2 predicted after 6 h. (**b**) The residual value of BDS-3 predicted after 6 h.

**Figure 11 sensors-23-02453-f011:**
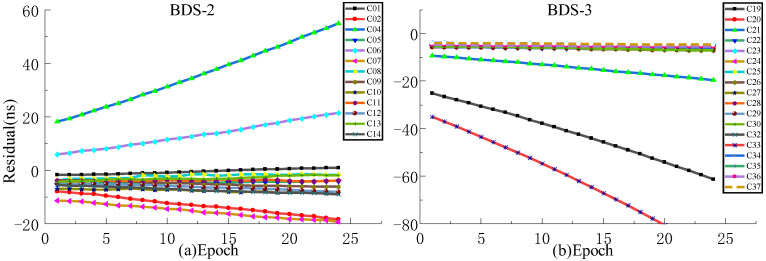
The residual plot of GM model using the initial hour update ultra-fast clock product of the next day. (**a**) The residual value of BDS-2 predicted after 6 h. (**b**) The residual value of BDS-3 predicted after 6 h.

**Figure 12 sensors-23-02453-f012:**
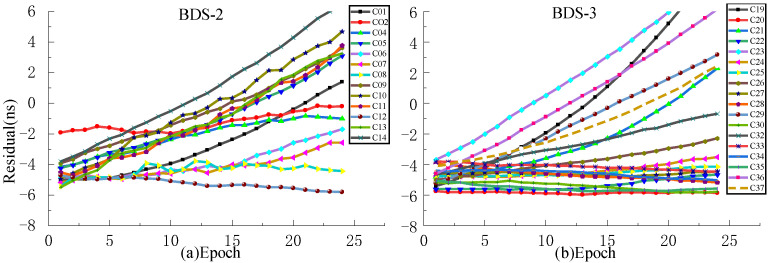
The residual plot of the SSA-ELM model using the initial hour update ultra-fast clock product of the next day. (**a**) The residual value of BDS-2 predicted after 6 h. (**b**) The residual value of BDS-3 predicted after 6 h.

**Figure 13 sensors-23-02453-f013:**
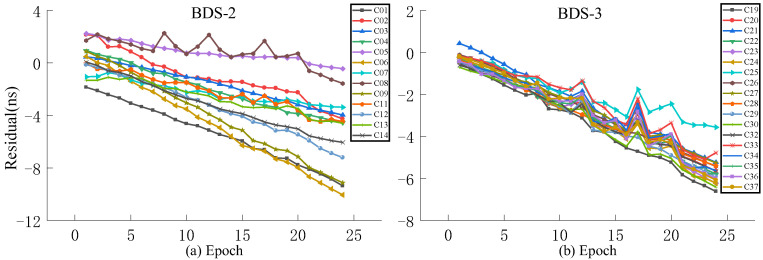
The residual plot of the ISUP model using the 6th hour update ultra-fast clock product of the day. (**a**) The residual value of BDS-2 predicted after 6 h. (**b**) The residual value of BDS-3 predicted after 6 h.

**Figure 14 sensors-23-02453-f014:**
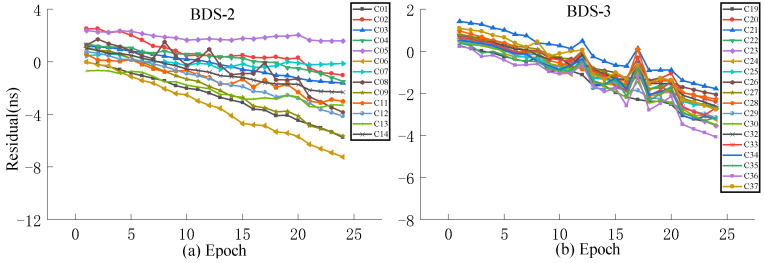
The residual plot of the QP model using the 6th hour update ultra-fast clock product of the day. (**a**) The residual value of BDS-2 predicted after 6 h. (**b**) The residual value of BDS-3 predicted after 6 h.

**Figure 15 sensors-23-02453-f015:**
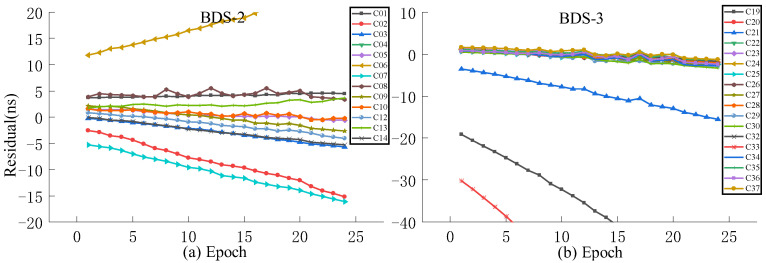
The residual plot of the GM model using the 6th hour update ultra-fast clock product of the day. (**a**) The residual value of BDS-2 predicted after 6 h. (**b**) The residual value of BDS-3 predicted after 6 h.

**Figure 16 sensors-23-02453-f016:**
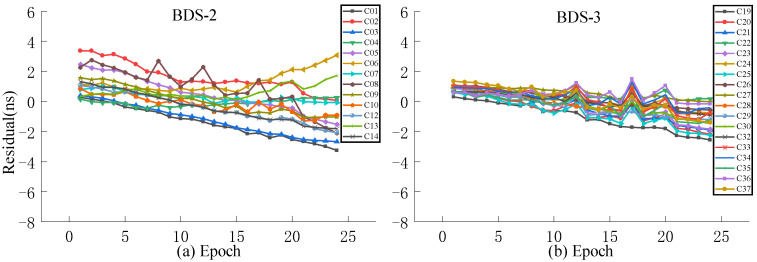
The residual plot of the SSA-ELM model using the 6th hour update ultra-fast clock product of the day. (**a**) The residual value of BDS-2 predicted after 6 h. (**b**) The residual value of BDS-3 predicted after 6 h.

**Figure 17 sensors-23-02453-f017:**
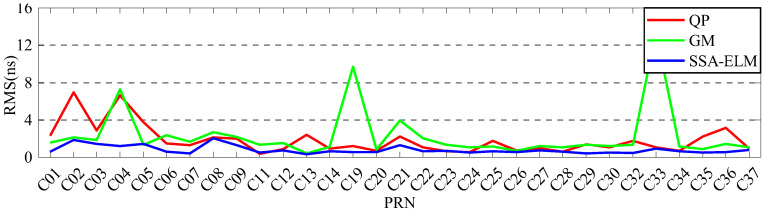
The prediction accuracies of BDS for 6 h obtained using 6 h of SCB data.

**Figure 18 sensors-23-02453-f018:**
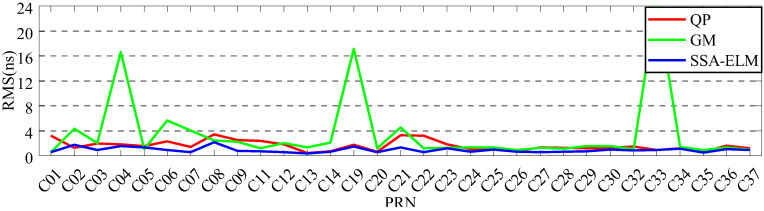
The prediction accuracies of BDS for 6 h obtained using 12 h of SCB data.

**Figure 19 sensors-23-02453-f019:**
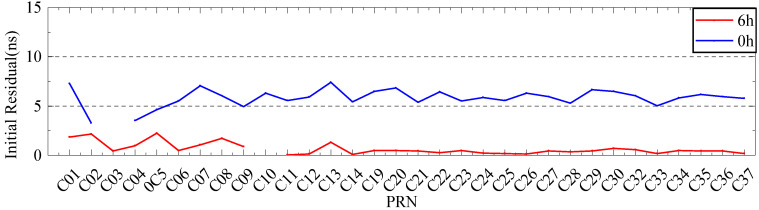
The initial residual value with a DOY of 196.

**Figure 20 sensors-23-02453-f020:**
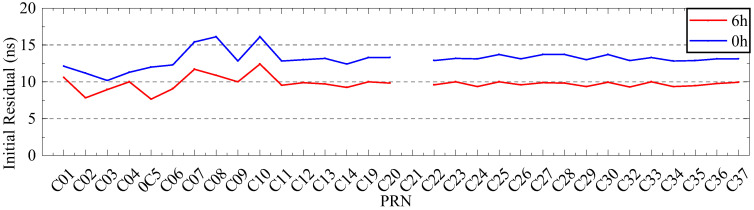
The initial residual value with a DOY of 198.

**Figure 21 sensors-23-02453-f021:**
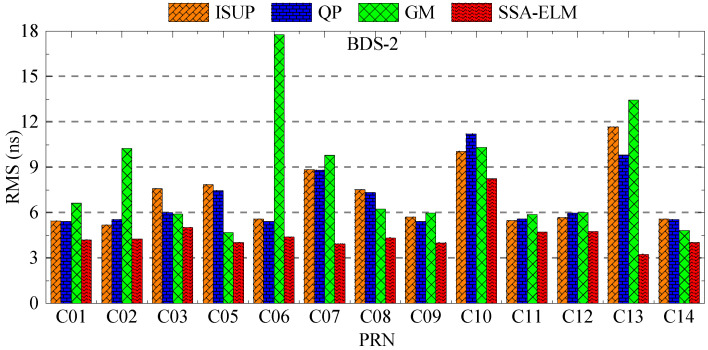
Multiday average prediction accuracy of BDS-2.

**Figure 22 sensors-23-02453-f022:**
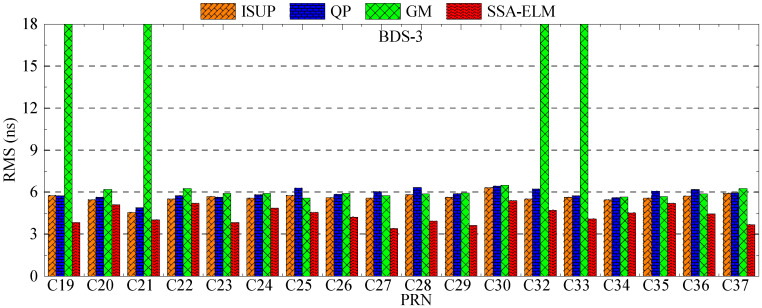
Multiday average prediction accuracy of BDS-3.

**Figure 23 sensors-23-02453-f023:**
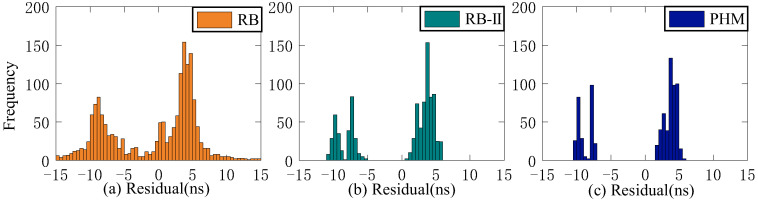
Multiday prediction residual statistics of the ISUP model. (**a**) The residual distribution of the RB clock. (**b**) The residual distribution of the Rb-II clock. (**c**) The residual distribution of the PHM clock.

**Figure 24 sensors-23-02453-f024:**
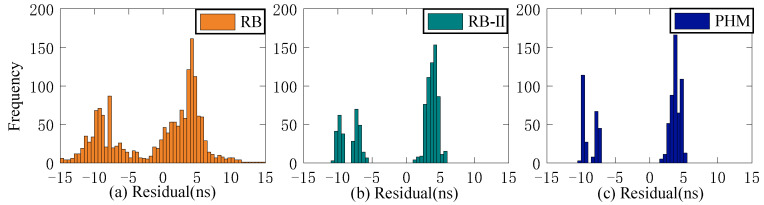
Multiday prediction residual statistics of the QP model. (**a**) The residual distribution of the RB clock. (**b**) The residual distribution of the Rb-II clock. (**c**) The residual distribution of the PHM clock.

**Figure 25 sensors-23-02453-f025:**
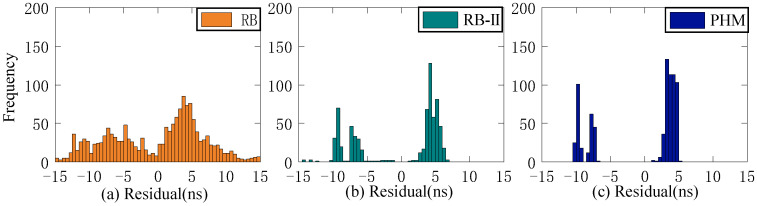
Multiday prediction residual statistics of the GM model. (**a**) The residual distribution of the RB clock. (**b**) The residual distribution of the Rb-II clock. (**c**) The residual distribution of the PHM clock.

**Figure 26 sensors-23-02453-f026:**
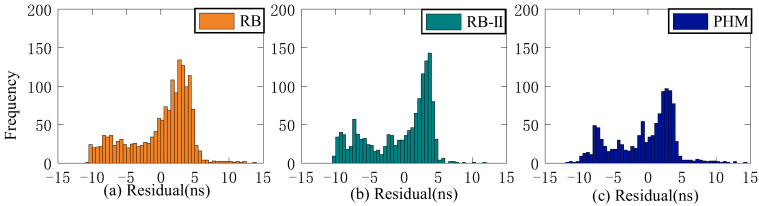
Multiday prediction residual statistics of the SSA-ELM model. (**a**) The residual distribution of the RB clock. (**b**) The residual distribution of the Rb-II clock. (**c**) The residual distribution of the PHM clock.

**Table 1 sensors-23-02453-t001:** Types of BDS satellite atomic clocks used in this work.

Satellite	Clock Type	PRN
BDS-2	Rb	C01, C02, C03, C04, C05, C06, C07, C08, C09, C10, C11, C12, C13, C14
BDS-3	Rb-II	C19, C20, C21, C22, C23, C24, C32, C33, C36, C37
	PHM	C25, C26, C27, C28, C29, C30, C34, C35

**Table 2 sensors-23-02453-t002:** The accuracy statistics of different reference clocks (ns).

Reference Clock	C11			C20			C27		
Second Difference	Rb	Rb-II	PHM	Rb	Rb-II	PHM	Rb	Rb-II	PHM
ISUP-ISC	2.68	6.00	4.75	6.41	0.92	1.97	4.30	1.89	0.87
ISUO-ISC	2.60	5.32	4.18	5.54	0.89	1.86	3.61	1.65	0.68
ISUP-ISUO	1.89	0.83	0.77	1.78	0.39	0.42	1.74	0.46	0.49
Mean	2.39	4.05	3.23	4.58	0.73	1.42	3.22	1.33	0.68

**Table 3 sensors-23-02453-t003:** The average accuracy statistics of different reference clocks for 13 days (ns).

Reference Clock	Rb			Rb-II			PHM		
Second Difference	Rb	Rb-II	PHM	Rb	Rb-II	PHM	Rb	Rb-II	PHM
ISUP-ISC	6.74	5.72	4.49	9.60	1.10	2.12	7.58	2.37	1.41
ISUO-ISC	5.32	5.33	4.27	8.08	0.70	1.59	6.36	1.72	0.80
ISUP-ISUO	3.57	3.02	2.71	3.60	0.93	1.15	3.04	1.15	1.05
Mean	5.21	4.69	3.82	7.10	0.91	1.62	5.66	1.75	1.09

**Table 4 sensors-23-02453-t004:** The average STD statistics of different atomic clocks (ns).

Second Difference	Rb	Rb-II	PHM	Mean
ISUP-ISC	3.71	1.41	0.99	2.03
ISUO-ISC	2.78	0.73	0.63	1.38
ISUP-ISUO	2.97	1.04	0.91	1.64
Mean	3.15	1.06	0.84	

**Table 5 sensors-23-02453-t005:** The average RMS values of 3 h for different prediction models based on 24 h of SCB data (ns).

Model	ISUP	QP	GM	SSA-ELM	Enhancement with ISUP (%)	Enhancement with QP (%)	Enhancement with GM (%)
Rb	1.75	1.09	2.88	0.97	44.67	11.58	66.32
Rb-II	1.65	0.56	1.37	0.55	66.57	1.99	59.85
PHM	1.62	0.50	0.50	0.49	70.01	2.81	3.20
Mean	1.67	0.72	1.58	0.67	60.42	5.46	57.59

**Table 6 sensors-23-02453-t006:** The average RMS values of 6 h for different prediction models based on 24 h of SCB data (ns).

Model	ISUP	QP	GM	SSA-ELM	Enhancement with ISUP (%)	Enhancement with QP (%)	Enhancement with GM (%)
Rb	3.19	1.88	3.93	1.13	64.66	39.93	71.25
Rb-II	3.35	1.48	2.09	0.80	76.04	45.71	61.72
PHM	3.19	1.48	1.27	0.76	76.10	48.31	40.05
Mean	3.25	1.61	2.43	0.90	72.27	44.65	62.96

**Table 7 sensors-23-02453-t007:** The average RMS values of 6 h for three prediction models based on 6 h of SCB data (ns).

Model	QP	GM	SSA-ELM	Enhancement with QP (%)	Enhancement with GM (%)
Rb	2.62	2.12	1.01	61.43	52.35
Rb-II	1.34	1.64	0.71	47.21	56.79
PHM	1.17	1.09	0.58	50.84	47.14
Mean	1.71	1.62	0.76	53.16	52.09

**Table 8 sensors-23-02453-t008:** The average RMS values of 6 h for three prediction models based on 12 h of SCB data (ns).

Model	QP	GM	SSA-ELM	Enhancement with QP (%)	Enhancement with GM (%)
Rb	1.89	2.40	0.97	48.33	59.40
Rb-II	1.68	1.60	0.93	44.56	41.46
PHM	1.08	1.24	0.77	29.08	38.28
Mean	1.55	1.75	0.89	40.66	46.38

**Table 9 sensors-23-02453-t009:** The statistics of the average RMS values of initial residuals in different clock bias files.

Clock Type	Rb	Rb-II	PHM	Mean
0h	9.75	9.42	9.64	9.60
6h	5.16	4.96	5.13	5.08

**Table 10 sensors-23-02453-t010:** The average RMS values of 6 days for three different prediction models (ns).

Model	ISUP	QP	GM	SSA-ELM	Enhancement with ISUP (%)	Enhancement with QP (%)	Enhancement with GM (%)
Rb	7.08	6.88	7.50	4.54	35.90	33.98	39.43
Rb-II	5.54	5.76	6.07	4.39	20.79	23.86	27.72
PHM	5.72	6.06	5.86	4.36	23.86	28.17	25.72
Mean	6.11	6.23	6.48	4.43	26.85	28.67	30.96

## Data Availability

In this paper, the datasets are from the sites: http://www.igmas.org/Product/Cpdetail/detail/nav_id/4/cate_id/36.html (accessed on 13 November 2022).

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
