# Peer review of "Improved BDS-2/3 Satellite Ultra-Fast Clock Bias Prediction Based with the SSA-ELM Model"

_sensors, 2023, doi:10.3390/s23052453_

Round 1

Reviewer 1 Report

1. IGS ultra-rapid is almost replaced by real-time service. IGS RTS broadcasts BDS SCB at ~s latency and with ~0.1-level ns precision. Why do you still study the prediction of IGSU products? you did not even mention RTS SCB in your introduction. That's the top concern for me.

2. the language must be polished. it's hard to follow the technicial meaning. sometimes, i guess... for example, double-difference? second difference? P1L26 'the prediction accuracy of SSA-ELM model is improved by...' basically you propose the SSA-ELM and should prove this model outperforms other model. but not the accuracy of the proposed model is improved. it should be 'the SSA-ELM model improves the prediction model by ....' In a word, the language prevents clear understanding of this submission. So far, i can only recommend a major concern based on current submission, and i hope that a simplified version would appear in the next review process.

3. This submission actually presents too many contents which distract readers from the main contribution. If the contribution is the SSA-ELM prediction model, why do you need Section 3 (almost five pages)? 

4. P3 Figure 1. there is no X.

5. Figure 2. The SSA box. No arrow between the 1st and 2nd rectangle. 

6. Figure 3 and below. What's the interval of one epoch? One hour???

7. Figure 9 and all others. bascially a clock jump should occur once the reference satellite is changed. that's the clock datum jump issue. but i did not see this datum jump issue throughout the submission. how do you deal with this issue?

Reviewer 2 Report

The whole work of "Improved BDS-2/3 satellite ultra-fast clock bias prediction based with SSA-ELM model" may be sufficient, but there are many mistakes and grammar errors.  The detail revise suggestion has been displayed in the attach file.

Reviewer 3 Report

This is an interesting manuscript presenting research on a new sparrow search algorithm for the optimisation of the extreme learning machine algorithm used for improving the performance of the satellite clock bias prediction. The authors have designed and applied their method to the clocks of BDS2/3 satellites. They confirmed already known better stability of the BDS3 clock versus BDS2 ones. They also found out that the prediction accuracy strongly depended on the selection of the reference clock type (Rb, PHM). They compared their results to the application of the existing prediction methods showing the advantage in terms of the resulting accuracy of their approach.

In my opinion, the manuscript is well-written, easy to follow, and the English language is clear. I have no major issues with this contribution, therefore my recommendation is to accept this work for publication with very minor revisions.

Minor comments: Tables 4-9: When talking about the accuracy, please state/clarify in the title that the tables present RMS or STD values.

Round 2

Reviewer 2 Report

Comments and Suggestions for Authors

1.      All the abbreviations should be interpreted at the first time, just as PHM.

2.      All the abbreviations “IGMAS” in the manuscript are not the Official description which should be “iGMAS”.

3.      As for learning Machine algorithm in the Figure 1, the node of output layer should be one, and how about the principle of determination the node of hidden layer?

4.      The full arc in line 104 should be ACs.

5.      Please uniform the format from line 202 to line 207.

6.      The English should be further improved, such as the precise real-time positioning which should be real-time precise point positioning. The English editing service should be adopted.

7.      Although, the file contains a few random corrections which caught my eyes while glancing over the text. Please check all changes for the correctness and make additional changes as needed.

8.      The point-by-point response to reviewers file should be upload with the revised manuscript. 

9.      The style of the references should be corrected by “Endnote”, as the format of the author should be two part but not three as in the journal of “Remote Sensing”.

Round 3

Reviewer 2 Report

The report are all reasonable, then the advice is to accept the manuscript.